# Physiological and Biochemical Responses to Salt Stress in Cultivated Eggplant (*Solanum melongena* L.) and in *S. insanum* L., a Close Wild Relative

**Marco Brenes** [1,2] **, Andrea Solana** [1] **, Monica Boscaiu** [3] **, Ana Fita** [1] **, Oscar Vicente** [1] **,**
**Ángeles Calatayud** [4] **, Jaime Prohens** [1] **and Mariola Plazas** [1,*]

[1] Institute for the Conservation and Improvement of Valencian Agrodiversity (COMAV), Universitat Politècnica de València, Camino de Vera 14, 46022 Valencia, Spain; marcob2103@gmail.com (M.B.); ansogar4@posgrado.upv.es (A.S.); anfifer@btc.upv.es (A.F.); ovicente@upvnet.upv.es (O.V.); jprohens@btc.upv.es (J.P.)

[2] Faculty of Biology, Instituto Tecnológico de Costa Rica, Avenida 14, calle 5, Cartago 30101, Costa Rica

[3] Mediterranean Agroforestry Institute (IAM), Universitat Politècnica de València, Camino de Vera 14, 46022 Valencia, Spain; mobosnea@eaf.upv.es

[4] Horticulture Department, Valencian Institute for Agriculture Research (IVIA), CV-315, Km 10.7, 46113 Moncada, Valencia, Spain; calatayud_ang@gva.es

* Correspondence: maplaav@btc.upv.es; Tel.: +34-96-387-9424

**Abstract:** Eggplant (*Solanum melongena*) has been described as moderately sensitive to salinity. We characterised the responses to salt stress of eggplant and *S. insanum*, its putative wild ancestor. Young plants of two accessions of both species were watered for 25 days with an irrigation solution containing NaCl at concentrations of 0 (control), 50, 100, 200, and 300 mM. Plant growth, photosynthetic activity, concentrations of photosynthetic pigments, $K^+$, $Na^+$, and $Cl^-$ ions, proline, total soluble sugars, malondialdehyde, total phenolics, and total flavonoids, as well as superoxide dismutase, catalase, and glutathione reductase specific activities, were quantified. Salt stress-induced reduction of growth was greater in *S. melongena* than in *S. insanum*. The photosynthetic activity decreased in both species, except for substomatal $CO_2$ concentration (Ci) in *S. insanum*, although the photosynthetic pigments were not degraded in the presence of NaCl. The levels of $Na^+$ and $Cl^-$ increased in roots and leaves with increasing NaCl doses, but leaf $K^+$ concentrations were maintained, indicating a relative stress tolerance in the two accessions, which also did not seem to suffer a remarkable degree of salt-induced oxidative stress. Our results suggest that the higher salt tolerance of *S. insanum* mostly lies in its ability to accumulate higher concentrations of proline and, to a lesser extent, $Na^+$ and $Cl^-$. The results obtained indicate that *S. insanum* is a good candidate for improving salt tolerance in eggplant through breeding and introgression programmes.

**Keywords:** eggplant; wild relative; vegetative growth; photosynthesis; ion homeostasis; osmolytes; oxidative stress

## 1. Introduction

Soil salinity affects over 1000 million ha of land throughout the world [1,2], and it continuously increases worldwide, affecting large areas of arable land [3]. The effects of soil salinity on plants vary depending on weather conditions, light intensity, soil characteristics, and species or taxonomic groups [4], but most crops are glycophytes and, therefore, are not able to grow on saline soils. Generally, growth of glycophytes is completely inhibited at salt concentrations in soil equivalent to 100–200 mM NaCl, eventually resulting in the death of the plant [5].

Eggplant (*Solanum melongena* L.) is one of the most popular vegetable crops throughout the world and, especially in Southeast Asia [6], and is moderately sensitive to salinity [7]. Eggplant fruits have a low calories content and contain high concentrations of phenolic acids, beneficial for human health [8,9]. Eggplant is cultivated on more than 1.86 million hectares and its annual production is over 54 million tonnes [6]. *Solanum melongena* can be crossed with a wide range of wild relatives from the primary, secondary, and tertiary genepools [10], and backcrossing to *S. melongena* of the interspecific hybrids for introgression breeding can result in the incorporation of traits from wild species into the eggplant genepool and in the broadening of the genetic basis of the crop [11–13]. Therefore, identifying sources of variation for tolerance to salinity among eggplant wild relatives, some of which grow in harsh environments, including areas prone to salinity [14], can contribute to breeding eggplant for higher tolerance to salinity. One of the most promising species for introgression breeding in eggplant is *S. insanum* L., which is the wild ancestor of eggplant and grows in a wide range of soil conditions [15]. Interspecific hybrids between *S. melongena* and *S. insanum* as well as backcrosses of the hybrids to *S. melongena*, are easily obtained and are highly fertile [10,11,16], which facilitates the transfer of traits from *S. insanum* to *S. melongena*.

To our knowledge, the responses of *S. insanum* under conditions of salt stress have not yet been studied. Data on physiological and biochemical traits under stressful conditions could be used as selection criteria for possible breeding programmes [17]. This study aims to determine the level of tolerance to salinity of *S. insanum*, comparing it to *S. melongena* by analysing the variation of growth traits, photosynthesis, and biochemical responses associated with tolerance to salinity, such as levels of ions accumulated in different tissues, osmolytes, and antioxidants. The results will provide relevant information on *S. insanum* as a possible source of variation of tolerance to salinity, for eggplant breeding.

## 2. Materials and Methods

### 2.1. Plant Material and Experimental Layout

The plant material used was provided by the Institute for the Conservation and Improvement of Valencian Agrodiversity (COMAV-UPV). *Solanum melongena* accession MEL1 originates from Ivory Coast, and *S. insanum* INS2 from Sri Lanka. *Solanum melongena* MEL1 was chosen as this accession is of particular interest for breeding as it has an excellent fruit set and shows a high degree of success in interspecific hybridisation [10,11]. Seeds were germinated following a shortened version of a protocol developed by Ranil et al. [18]. Briefly, seeds were soaked first for 24 h in water and for an additional 24 h in a 500 ppm solution of gibberellic acid ($GA_3$), and then placed in Petri dishes on filter paper moistened with a solution of 1000 ppm $KNO_3$ and subjected to a heat shock treatment at 37 °C for 24 h. The Petri dishes were transferred to a growth chamber under conditions of 16 h light/8 h darkness at 25 °C until germination was completed. Once germinated, seedlings were placed in seedbeds and kept under the same conditions of light and temperature for two weeks. Seedlings homogenous in size were selected and transplanted to small pots and, subsequently, to 1.3 L pots with 500 g of Huminsubstrat N3 (Klasmann-Deilmann, Geeste, Germany) commercial substrate. The plants were transferred to a greenhouse with benches and controlled temperature (maximum of 30 °C and minimum of 15 °C) for acclimatisation for 20 days, and when plants developed 6–8 fully expanded leaves, the stress treatments were started. Five plants of each species, each one corresponding to a biological replica, were irrigated every four days with 1.25 L of NaCl solutions (final concentrations: 50, 100, 200, and 300 mM NaCl dissolved in deionised water) or deionised water for the control plants, for 25 days, and several non-destructive growth parameters were measured in all plants (stem length, stem diameter, and number of leaves). Runoff water after irrigation was allowed to freely drain. Measurements for physiological, biochemical, and ion content parameters were based on one technical replicate.

## 2.2. Electrical Conductivity of the Substrate

Electrical conductivity of the substrate was measured in a 1:5 suspension ($EC_{1:5}$). At the end of the treatments, after removing the plants from the pots, the remaining substrate was dried in an oven at 65 °C for four days and a soil/water (1:5) suspension was prepared in deionised water and stirred for 1 h at 600 rpm and 21 °C. EC was measured with a Crison Conductivity-meter 522 (Crison Instruments SA, Barcelona, Spain) and expressed in dS m$^{-1}$.

## 2.3. Gaseous Exchange

At the end of the stress period (25 days), the $CO_2$ assimilation rate ($A_N$, $\mu$mol $CO_2$ m$^{-2}$ s$^{-1}$), stomatal conductance to water vapor (gs, mol $H_2O$ m$^{-2}$ s$^{-1}$), substomatal $CO_2$ concentration (Ci, $\mu$mol $CO_2$ mol$^{-1}$ air), and transpiration rate (E, mmol $H_2O$ m$^{-2}$ s$^{-1}$) were measured in one of the fully developed leaves of each plant using a portable LI-COR 6400 infrared gas analyser (Li-Cor Inc., Lincoln, NE, USA).

## 2.4. Evaluation of Growth Parameters

To assess the effect of salt stress on the two species, several growth parameters were analysed at the end of the treatments: fresh weight of roots (RFW), stems (SFW), and leaves (LFW); length of roots (RL) and stems (SL); stem diameter (SD); and area of the largest leaf (LA). Stem elongation (SE), stem thickening (ST), and increase in the number of leaves (Lno) were calculated as the difference between the final and initial values of stem length, stem diameter, and number of leaves, respectively, in the same plant. The water content of roots (RWC), stems (SWC), and leaves (LWC) was determined by weighing a part of fresh material, drying it for four days at 60 °C, and weighing it again; the humidity percentage was calculated with the following formula: [(Fresh weight − Dry weight)/Fresh weight] * 100.

## 2.5. Ion Quantification

Contents of potassium ($K^+$), sodium ($Na^+$), and chloride ($Cl^-$) were determined in roots and leaves. Samples of 50 mg of ground dry plant material in 15 mL of deionised water were heated at 95 °C for one hour, followed by cooling on ice and filtration through a 0.45 $\mu$m nylon filter [19]. The $Na^+$ and $K^+$ content was quantified with a PFP7 flame photometer (Jenway Inc., Burlington, VT, USA), and the $Cl^-$ content was determined using a chlorimeter (Sherwood, model 926, Cambridge, UK).

## 2.6. Quantification of Photosynthetic Pigments

The content of chlorophyll a (Chl a), chlorophyll b (Chl b), and carotenoids (Caro) was determined using the methodology described by Lichtenthaler and Wellburn [20]. Pigments were extracted from 50 mg fresh plant material using 10 mL of ice-cold 80% acetone (v/v), and the extracts were diluted 10 times using the same solvent. The absorbance was measured at 470, 645, and 663 nm ($A_{470}$, $A_{645}$, and $A_{663}$, respectively), and the following formulas were used to calculate the different pigments:

$$\text{Chl a } (\mu\text{g mL}^{-1}) = 12.21 \times A_{663} - 2.81 \times A_{646} \tag{1}$$

$$\text{Chl b } (\mu\text{g mL}^{\pm1}) = 20.13 \times A_{646} - 5.03 \times A_{663} \tag{2}$$

$$\text{Caro } (\mu\text{g mL}^{-1}) = (1000 \times A_{470} - 3.27 \times [\text{Chl a}] - 104 \times [\text{Chl b}])/227 \tag{3}$$

## 2.7. Quantification of Osmolytes

The quantification of free proline (Pro) was carried out following the acetic acid-ninhydrin method [21]. An aqueous solution (2 mL) of 3% (w/v) sulfosalicylic acid was added to 50 mg freshly ground plant material (from each biological replica). One volume of extract was mixed with one

volume of ninhydrin acid and one volume of glacial acetic acid, and then the mix was placed in a water bath at 95 °C for one hour, and subsequently cooled for 10 min on ice and extracted with toluene. The absorbance of the organic phase was determined at 520 nm using toluene as the blank.

Total soluble sugars (TSSs) were measured according to the methodology described in [22]. Fresh leaf material (50 mg) was ground and mixed with 3 mL of 80% (v/v) methanol on a rocker shaker for 24 h, and the extract was recovered by centrifugation; concentrated sulfuric acid and 5% phenol were added to the supernatant and the absorbance was measured at 490 nm. TSS contents were expressed as 'mg equivalent of glucose' per g dry weight (DW).

### 2.8. Measurement of Malondialdehyde (MDA) and Antioxidant Compounds

MDA, total phenolic compounds (TPCs), and total flavonoids (TFs) were measured in plant extracts prepared from 50 mg ground fresh leaf material using 80% (v/v) methanol. For MDA quantification, extracts were mixed with 0.5% thiobarbituric acid (TBA) prepared in 20% trichloroacetic acid (TCA), or with 20% TCA without TBA for the controls, and then incubated at 95°C for 20 min, cooled on ice, and centrifuged at 12,000× $g$ for 10 min at 4 °C [23]. The absorbance of the supernatants was measured at 532 nm. The non-specific absorbance at 600 and 440 nm was subtracted, and MDA concentration was determined using the equations included in [23], based on the extinction coefficient of the MDA-TBA adduct at 532 nm. The concentration of MDA was expressed as nmol $g^{-1}$ DW.

TPCs were measured using the Folin–Ciocalteu reagent [24]. Methanol extracts were mixed with $Na_2CO_3$ and the reagent and, after 90 min of incubation in the dark, the absorbance was measured at 765 nm. A standard reaction was performed in parallel using known amounts of gallic acid (GA), and TPC contents were reported as equivalents of GA (mg eq. GA $g^{-1}$ DW).

Total flavonoids (TFs) were quantified according to the method described by Zhisen et al. [25], based on the nitration of aromatic rings containing a catechol group. Methanol extracts of each sample were reacted with $NaNO_2$ and $AlCl_3$ under alkaline conditions, and the absorbance at 510 nm was measured. The concentration of TFs was expressed as equivalents of catechin, used as the standard (mg eq. C $g^{-1}$ DW).

### 2.9. Antioxidant Enzyme Activities

The activities of superoxide dismutase (SOD), catalase (CAT), and glutathione reductase (GR) were measured in crude protein extracts prepared from frozen (−70 °C) leaf material, as previously described [26]. Enzyme activities in the extracts were expressed as 'specific activities', in units per mg of protein.

SOD activity in the protein extracts was determined as described by Beyer and Fridovich [27], following the inhibition of nitroblue tetrazolium (NBT) photoreduction by measuring the absorbance of the sample at 560 nm. The reaction mixtures contained riboflavin as the source of superoxide radicals. One SOD unit was defined as the amount of enzyme causing 50% inhibition of NBT photoreduction under the assay conditions.

CAT activity was measured by the decrease in absorbance at 240 nm, which accompanies the consumption of $H_2O_2$ added to protein extracts [28]. One CAT unit was defined as the amount of enzyme that will decompose one mmol of $H_2O_2$ per minute at 25 °C.

The protocol of Conell and Mullet [29] was used for the GR assays, following the oxidation of NADPH (the cofactor in the GR-catalysed reduction of oxidised glutathione (GSSG)) by the decrease in absorbance at 340 nm. One GR unit was defined as the amount of enzyme that will oxidise one mmol of NADPH per minute at 25 °C.

### 2.10. Statistical Analysis

Data were analysed using the software Statgraphics Centurion v. XVI (Statpoint Technologies Inc., Warrenton, VA, USA). The significance of the differences between treatments (for each species), between species (for each treatment) and their interaction were evaluated through a two-factorial analysis

of variance (ANOVA) for traits related to plant growth, photosynthetic pigments, photosynthesis parameters, osmolytes, MDA, and antioxidants. For ion accumulation, an additional factor (organ) was included and a three-way factor analysis of variance (ANOVA) (treatment, species, and organ) was performed. Post-hoc comparisons were made using the Tukey Honestly Significant Difference (HSD) test at $p < 0.05$ for the effects of treatment within species (and combinations of species and organ in the case of ions). All the parameters measured in plants of the control and salt stress treatments were subjected to multivariate analysis through a principal component analysis (PCA).

## 3. Results

### 3.1. Substrate Electrical Conductivity

Electrical conductivity of the substrate increased in parallel to the concentration of NaCl, applied in a similar manner in both species, as indicated by the analysis of variance, which detected significant differences only between treatments, but not between the two species. EC reached the highest levels at the end of the treatments (19.19 dS m$^{-1}$ for *S. melongena* and 23.66 dS m$^{-1}$ for S. *insanum*) in the pots watered with 300 mM NaCl (Figure 1).

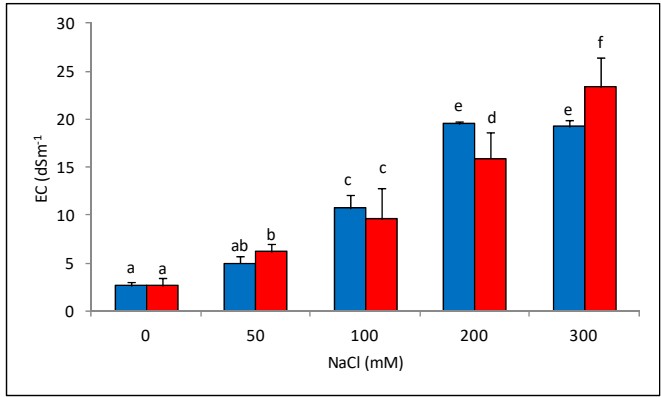

**Figure 1.** Electrical conductivity (EC$_{1:5}$) of the pot substrates after 25 days of treatment with the indicated NaCl concentrations, in *Solanum melongena* (blue) and *S. insanum* (red). Same letters indicate homogeneous groups between combinations of treatments for EC according to the Tukey test ($p < 0.05$, $n = 5$).

### 3.2. Analysis of Morphological and Photosynthetic Parameters

Salt stress inhibited the growth of the two species, in a concentration-dependent manner. Several growth parameters were determined in control and salt-stressed plants, at the end of the treatments, and a two-way ANOVA was performed, considering the effect of treatment, species, and their interaction (Table 1). The effect of 'species' was significant for most of the parameters, except root length (RL), some stem traits [stem elongation (SE), thickening (ST), fresh weight (SFW), water content (SWC)], total fresh weight (TFW), and chlorophyll a (Chl a). The effect of 'treatment' was significant for all traits analysed, except water content of roots (RWC), stems (SWC), and leaves (LWC), as well chlorophylls a and b (Chl a and Chl b). The interaction of the two factors was significant only for stem elongation (SE), the increase in leaf number (Lno), leaf fresh weight (LFW), and the area of the largest leaf (Table 1).

**Table 1.** Two-way analysis of variance (ANOVA) of species, treatment, and their interactions, for the indicated parameters. Numbers shown represent percentages of the sum of squares (SS).

|  | Abbr. | Treatment [a] | Species [a] | Interaction [a] | Residual |
|---|---|---|---|---|---|
| Root length | RL | 11.42 * | 1.32 | 20.77 | 66.48 |
| Root fresh weight | RFW | 6.36 | 24.43 *** | 2.60 | 66.60 |
| Root water content | RWC | 55.17 *** | 14.41 *** | 5.12 | 25.29 |
| Stem elongation | SE | 68.59 *** | 2.26 | 6.44 * | 22.71 |

**Table 1.** *Cont.*

|  | Abbr. | Treatment [a] | Species [a] | Interaction [a] | Residual |
|---|---|---|---|---|---|
| Stem thickening | ST | 63.70 *** | 0.93 | 2.32 | 32.96 |
| Stem fresh weight | SFW | 56.05 *** | 0.26 | 3.30 | 40.54 |
| Stem water content | SWC | 5.04 | 4.83 | 4.66 | 85.47 |
| Increase in no. of leaves | Lno | 34.91 *** | 29.19 *** | 10.20 ** | 39.82 |
| Leaf area | LA | 34.69 *** | 29.19 *** | 10.20 ** | 25.69 |
| Leaf fresh weight | LFW | 43.15 *** | 10.54 ** | 11.11 * | 35.20 |
| Leaf water content | LWC | 7.45 | 7.10 * | 17.34 | 68.11 |
| Total fresh weight | TFW | 44.77 *** | 0.15 | 5.98 | 49.09 |
| Chlorophyll a | Chl a | 14.09 | 3.33 | 4.58 | 78.00 |
| Chlorophyll b | Chl b | 12.86 | 8.37 * | 9.39 | 69.43 |
| Carotenoids | Caro | 18.92 * | 13.35 *** | 4.19 | 63.40 |
| Photosynthestic rate | $A_N$ | 23.29 *** | 37.82 *** | 4.75 | 34.58 |
| Stomatal conductance | gs | 16.22 * | 29.92 *** | 3.63 | 50.23 |
| Int. $CO_2$ concentration | Ci | 29.10 ** | 17.22 *** | 1.11 | 52.57 |
| Transpiration rate | E | 17.92 ** | 39.17 ** | 2.35 | 40.56 |

[a] ***, **, and * indicate significant at $p < 0.001$, $p < 0.01$, and $p < 0.05$, respectively.

At the root level, the effect of salt was more pronounced in *S. melongena*, as root length (RL) and root fresh weight (RFW) did not vary significantly in *S. insanum* (Table 2). In both species, the water content of the roots increased with salinity. Growth of the stems was affected by salinity, but the water content was maintained stable in both species. Regarding the analysed leaf parameters, all showed a significant decrease in salt-treated plants of *S. melongena*, whereas in S. *insanum*, their variation was not significant, except for the increase in the number of leaves (Lno). When considering the total fresh weight (TFW), the reduction was significant only in the cultivated eggplant, but not in the wild species, in which, at the lowest concentration applied, TFW even increased, although the variation was not statistically significant in relation to the control. In both species, the water content of the leaves (LWC) did not vary significantly with the treatments (Table 2). Moreover, the variation between treatments of Chl a and Chl b was non-significant, whereas carotenoids decreased only in *S. insanum*. Stomatal conductance (gs), internal concentration of $CO_2$ (Ci), and transpiration (E) decreased in *S. melongena*, but not in *S. insanum*; photosynthesis rate ($A_N$), on the other hand, showed a significant reduction in both species (Table 2).

**Table 2.** Growth responses and photosynthetic parameters in *Solanum melongena* (MEL) and *S. insanum* (INS) after 25 days of treatment with the indicated NaCl concentrations.

| Trait | Taxa | Treatment (mM NaCl) | | | | |
|---|---|---|---|---|---|---|
|  |  | 0 | 50 | 100 | 200 | 300 |
| RL | MEL | 26.2 ± 1.3 [c] | 26.7 ± 2.2 [c] | 24.6 ± 1.7 [bc] | 19.8 ± 1.7 [ab] | 17.8 ± 0.6 [a] |
|  | INS | 21.0 ± 3.5 [A] | 20.2 ± 0.5 [A] | 24.0 ± 1.1 [A] | 20.1 ± 2.5 [A] | 24.2 ± 2.9 [A] |
| RFW | MEL | 9.0 ± 0.8 [b] | 9.8 ± 0.5 [b] | 9.1 ± 0.7 [b] | 9.1 ± 0.5 [b] | 6.9 ± 0.1 [a] |
|  | INS | 10.8 ± 2.1 [A] | 11.9 ± 0.8 [A] | 11.2 ± 0.3 [A] | 11.4 ± 0.9 [A] | 10.9 ± 0.8 [A] |
| RWC | MEL | 71.2 ± 1.2 [a] | 76.8 ± 0.7 [b] | 78.8 ± 0.7 [bc] | 80.4 ± 0.4 [c] | 79.5 ± 0.5 [bc] |
|  | INS | 65.5 ± 3.6 [A] | 68.4 ± 1.2 [AB] | 76.0 ± 1.0 [BC] | 78.0 ± 0.5 [C] | 77.7 ± 0.6 [C] |
| SE | MEL | 6.3 ± 1.9 [c] | 5.7 ± 1.1 [bc] | 5.8 ± 0.5 [c] | 3.7 ± 0.5 [ab] | 2.5 ± 0.2 [a] |
|  | INS | 8.3 ± 0.6 [C] | 6.7 ± 0.2 [C] | 4.9 ± 0.8 [BC] | 3.7 ± 0.2 [AB] | 3.3 ± 0.4 [A] |
| ST | MEL | 3.5 ± 0.2 [c] | 2.9 ± 0.2 [bc] | 2.3 ± 0.1 [bc] | 1.6 ± 0.3 [ab] | 1.3 ± 0.3 [a] |
|  | INS | 3.2 ± 0.5 [B] | 2.2 ± 0.4 [AB] | 2.2 ± 0.3 [AB] | 1.8 ± 0.2 [AB] | 1.1 ± 0.1 [A] |
| SFW | MEL | 4.4 ± 0.9 [b] | 4.7 ± 0.2 [b] | 3.9 ± 0.5 [ab] | 2.6 ± 0.2 [ab] | 1.9 ± 0.3 [a] |
|  | INS | 5.5 ± 1.0 [C] | 4.7 ± 0.2 [BC] | 3.5 ± 0.1 [B] | 2.5 ± 0.1 [AB] | 2.1 ± 0.1 [A] |
| SWC | MEL | 68.4 ± 6.2 [a] | 78.1 ± 0.7 [a] | 76.8 ± 3.9 [a] | 73.6 ± 3.5 [a] | 75.1 ± 5.6 [a] |
|  | INS | 70.8 ± 2.3 [A] | 70.6 ± 1.1 [A] | 72.8 ± 1.7 [A] | 71.4 ± 0.7 [A] | 67.6 ± 1.4 [A] |

**Table 2.** *Cont.*

| Trait | Taxa | Treatment (mM NaCl) | | | | |
|---|---|---|---|---|---|---|
| | | 0 | 50 | 100 | 200 | 300 |
| Lno | MEL | 2.4 ± 0.2 [ab] | 2.6 ± 0.2 [ab] | 3.0 ± 0.0 [b] | 1.8 ± 0.2 [a] | 1.8 ± 0.2 [a] |
| | INS | 2.2 ± 0.4 [C] | 1.8 ± 0.4 [BC] | 1.4 ± 0.2 [B] | 0.8 ± 0.4 [AB] | −0.4 ± 0.2 [A] |
| LA | MEL | 205.7 ± 12.0 [c] | 161.6 ± 8.4 [b] | 149.2 ± 4.1 [ab] | 132.0 ± 3.9 [ab] | 119.4 ± 8.6 [a] |
| | INS | 143.4 ± 17.0 [A] | 148.6 ± 3.7 [A] | 139.8 ± 8.2 [A] | 136.6 ± 8.2 [A] | 105.5 ± 9.1 [A] |
| LFW | MEL | 22.4 ± 1.9 [d] | 19.3 ± 1.0 [cd] | 16.6 ± 0.8 [bc] | 12.9 ± 1.1 [b] | 9.1 ± 0.7 [a] |
| | INS | 13.8 ± 3.2 [A] | 14.8 ± 1.1 [A] | 14.4 ± 0.7 [A] | 12.0 ± 0.3 [A] | 9.6 ± 1.1 [A] |
| LWC | MEL | 81.3 ± 2.3 [a] | 84.4 ± 4.8 [a] | 85.3 ± 1.7 [a] | 81.9 ± 2.7 [a] | 72.2 ± 2.8 [a] |
| | INS | 75.5 ± 4.5 [A] | 78.3 ± 0.8 [A] | 79.5 ± 0.7 [A] | 80.8 ± 0.3 [A] | 82.2 ± 0.7 [A] |
| TFW | MEL | 35.8 ± 2.8 [d] | 33.8 ± 1.5 [cd] | 29.7 ± 3.2 [c] | 24.5 ±0.8 [b] | 18.0 ± 1.3 [a] |
| | INS | 30.1 ± 6.0 [a] | 31.5 ± 1.9 [a] | 29.1 ± 2.1 [a] | 25.9 ± 0.9 [a] | 22.7 ± 1.8 [a] |
| Chl a | MEL | 9.4 ± 2.2 [a] | 8.4 ± 1.1 [a] | 11.0 ± 1.1 [a] | 5.7 ± 1.1 [a] | 8.1 ± 0.7 [a] |
| | INS | 7.6 ± 1.9 [A] | 6.8 ± 1.9 [A] | 8.7 ± 1.1 [A] | 7.2 ± 0.5 [A] | 6.5 ± 0.5 [A] |
| Chl b | MEL | 4.3 ± 1.0 [a] | 3.8 ± 0.6 [a] | 5.7 ± 1.4 [a] | 2.2 ± 0.3 [a] | 3.6 ± 0.4 [a] |
| | INS | 3.3 ± 0.8 [A] | 2.5 ± 0.7 [A] | 2.9 ± 0.4 [A] | 2.7 ± 0.8 [A] | 3.1 ± 0.3 [A] |
| Caro | MEL | 1.6 ± 0.4 [a] | 1.3 ± 0.5 [a] | 1.3 ± 0.4 [a] | 0.9 ± 0.4 [a] | 1.3 ± 0.3 [a] |
| | INS | 2.1 ± 0.3 [B] | 1.9 ± 0.2 [AB] | 1.7 ± 0.2 [AB] | 1.4 ± 0.1 [AB] | 1.2 ± 0.1 [A] |
| $A_N$ | MEL | 8.5 ± 1.1 [b] | 9.9 ± 1.3 [b] | 9.0 ± 0.8 [b] | 5.4 ± 0.9 [ab] | 3.4 ± 0.3 [a] |
| | INS | 18.9 ± 3.3 [B] | 15.4 ± 1.5 [AB] | 18.2 ± 2.0 [AB] | 15.8 ± 1.9 [AB] | 7.7 ± 2.9 [A] |
| gs | MEL | 0.11 ± 0.00 [b] | 0.11 ± 0.00 [b] | 0.09 ± 0.01 [ab] | 0.05 ± 0.00 [ab] | 0.03 ± 0.00 [a] |
| | INS | 0.24 ± 0.10 [A] | 0.26 ± 0.10 [A] | 0.30 ± 0.15 [A] | 0.21 ± 0.10 [A] | 0.09 ± 0.03 [A] |
| C | MEL | 234.6 ± 17.0 [b] | 230.0 ± 8.5 [b] | 213.8 ± 5.6 [b] | 195.0 ± 5.7 [ab] | 182.2 ± 9.7 [a] |
| | INS | 274.6 ± 24.0 [A] | 248.2 ± 13.0 [A] | 243.2 ± 13.0 [A] | 222.0 ± 14.0 [A] | 219.2 ± 5.3 [A] |
| E | MEL | 2.6 ± 0.5 [b] | 2.7 ± 0.4 [b] | 2.5 ± 0.1 [b] | 1.5 ± 0.2 [ab] | 0.9 ± 0.1a |
| | INS | 4.7 ± 1.0 [A] | 5.3 ± 0.8 [A] | 5.8 ± 0.8 [A] | 4.8 ± 0.8 [A] | 2.6 ± 0.4 [A] |

Mean ± SE values are shown (*n* = 5). Same letters within each row (lowercase for *S. melongena* and capital letters for *S. insanum*) indicate homogeneous groups between treatments for each species, according to the Tukey HSD test ($p < 0.05$). Abbreviations: root length (RL; cm), root fresh weight (RFW; g), root water content (RWC; %), stem elongation (SE; cm), stem thickening (ST; mm), stem fresh weight (SFW; g), stem water content (SWC, %), increase in the number of leaves (Lno), area of the largest leaf (LA; cm$^2$), leaf fresh weight (LFW; g), leaf water content (LWC; %), total fresh weight (TFW; g), chlorophyll a (Chl a; mg g$^{-1}$ dry weight (DW)), chlorophyll b (Chl b; mg g$^{-1}$ DW), carotenoids (Caro; mg g$^{-1}$ DW), photosynthestic rate ($A_N$; μmol $CO_2$ m$^{-2}$ s$^{-1}$), stomatal conductance (gs; mol $H_2O$ m$^{-2}$ s$^{-1}$), internal concentration of $CO_2$ (Ci; μmol $CO_2$ mol$^{-1}$ air), and transpiration rate (E; mmol $H_2O$ m$^{-2}$ s$^{-1}$).

For an easier estimation of the pattern of variation of growth parameters in the two species, the variation of fresh weight and water content in the roots, stems, and leaves of the plants subjected to the salt treatments is shown in Figure 2, as percentages of the values measured in the corresponding non-stressed controls. In general, both fresh weight (FW) and water content (WC) showed a relatively smaller reduction in *S. insanum* than in *S. melongena*, at least in roots and leaves, and more pronounced at the highest salt concentration tested (Figure 2).

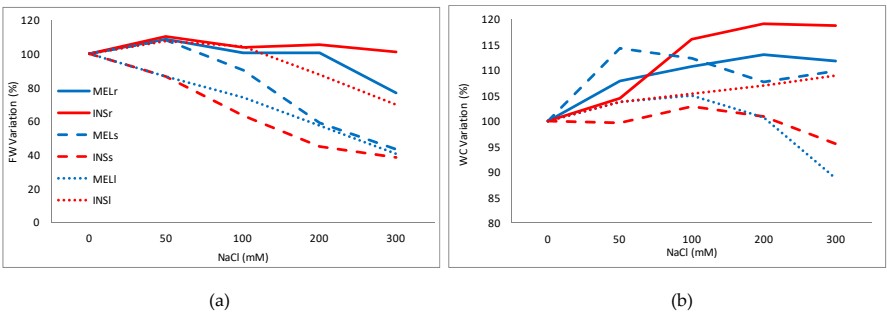

(a)                (b)

**Figure 2.** Reduction of fresh weight (FW) (**a**) and water content (WC) (**b**) in roots (MELr and INSr), stems (MELs and INSs), and leaves (MELl and INSl) of *Solanum melongena* (MEL; blue lines) and *S. insanum* (INS; red lines) plants after 25 days of salt treatments at the indicated NaCl concentrations. Values are shown as percentages of the corresponding controls (0 mM NaCl).

### 3.3. Ion Accumulation

To analyse the changes in ion contents in the plants, in response to the salt treatments, a multifactorial ANOVA was performed, considering the effect of the treatment, species, organs of the plants (roots vs. leaves), and their interactions (Table 3). In the case of $Na^+$ and $Cl^-$ contents and the $K^+/Na^+$ ratio, the main effect was that of the treatment, which was highly significant for all traits, whereas the 'species' factor was significant only for $Cl^-$ and $K^+$. The effect of the 'organ' variable was significant for $Cl^-$, $K^+$, and the $K^+/Na^+$ ratio, but it was by far the greatest contributor to the sums of squares for $K^+$, as leaves of both species contain considerably higher concentrations of $K^+$ than the roots. Some significant double and triple interactions were detected, for example, between 'treatment' and 'species' or between 'treatment' and 'organ' for $Na^+$ and $Cl^-$, but their contribution to the sums of squares was generally low (below 3.5%), except for the interaction between 'treatment' and 'organ' for the $K^+/Na^+$ ratio (Table 3).

**Table 3.** Factorial analysis of variance (ANOVA) considering the effect of treatment (A), species (B), organ (C), and their interactions (A × B; A × C; B × C; A × B × C) on ions ($Na^+$, $Cl^-$, $K^+$) contents and the $K^+/Na^+$ ratio, in *Solanum melongena* and *S. insanum*. Numbers represent percentages of sum of squares (SS).

| Ion Contents and $K^+/Na^+$ Ratio | A [a] | B [a] | C [a] | AB [a] | AC [a] | BC [a] | ABC [a] | Residuals |
|---|---|---|---|---|---|---|---|---|
| $Na^+$ | 84.70 *** | 0.24 | 0.04 | 2.00 ** | 1.16 * | 1.2 ** | 1.45 ** | 8.61 |
| $Cl^-$ | 79.30 *** | 1.85 *** | 1.81 *** | 2.52 ** | 1.29 * | 1.01 * | 2.13 * | 10.07 |
| $K^+$ | 2.30 *** | 6.10 *** | 70.20 *** | 0.58 | 2.14 * | 0.61 | 3.39 ** | 14.69 |
| $K^+/Na^+$ | 71.27 *** | 0.04 | 15.54 *** | 0.25 | 10.42 *** | 0.01 | 0.16 | 2.26 |

[a] ***, **, and * indicate significant at $p < 0.001$, $p < 0.01$, and $p < 0.05$, respectively.

In both species, $Na^+$ and $Cl^-$ concentrations increased in parallel to the increase in external salinity, in the roots and the leaves of the plants (Figure 3a,b). The pattern of variation was similar in the two species, as were, in general, the contents of both ions in roots and leaves for each NaCl concentration tested, except that *S. insanum* accumulated higher levels of $Na^+$ and $Cl^-$ in leaves than in roots at high salinity (200–300 mM NaCl). On the contrary, $K^+$ levels remained generally steady in response to the salt treatments, in roots and leaves of the two species, and in all cases, significantly higher in the leaves (Figure 3c). The salt-induced increase in $Na^+$ concentrations, accompanied by no significant changes of $K^+$ contents, led to a significant decrease of the $K^+/Na^+$ ratio in both species, especially in the leaves, where the initial values in the controls were higher than in roots (Figure 3d).

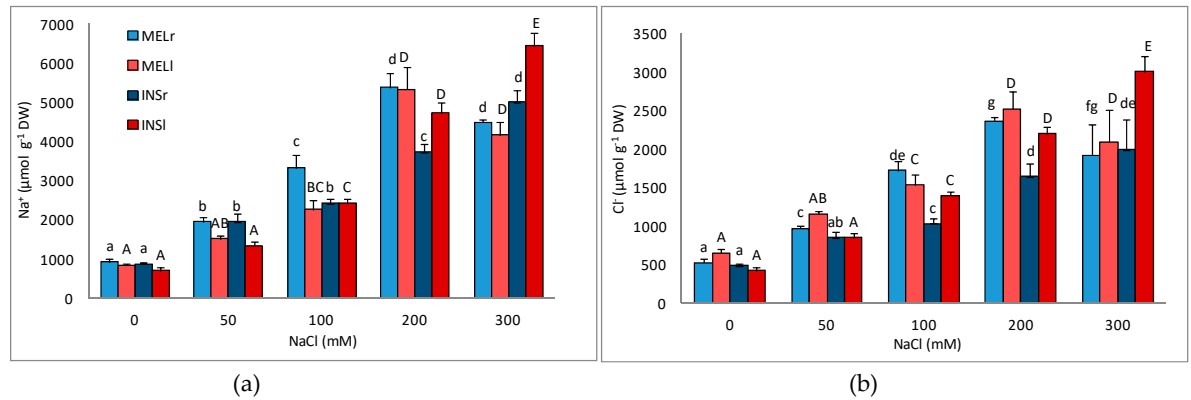

(a)　　　　　　　　　　　　　　　　　　　(b)

**Figure 3.** *Cont.*

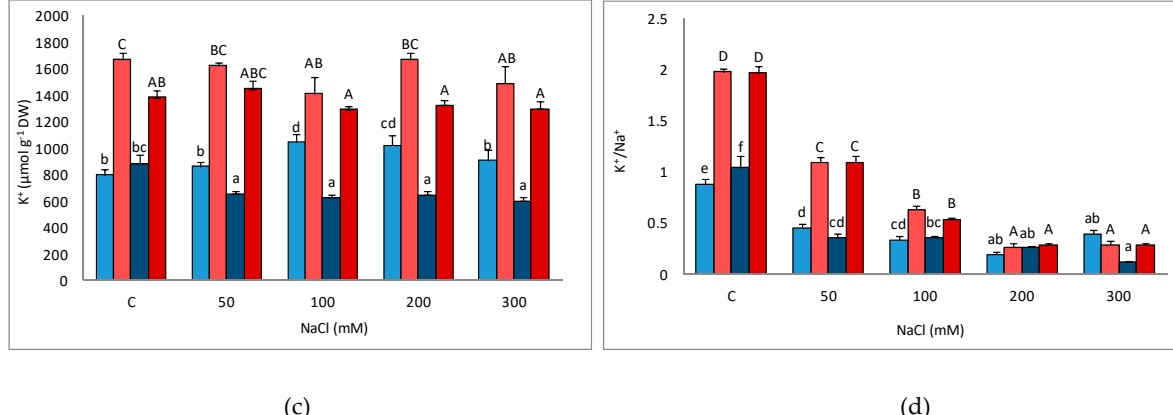

<center>(c)                    (d)</center>

**Figure 3.** $Na^+$ (**a**), $Cl^-$ (**b**), and $K^+$ (**c**) contents and $K^+/Na^+$ ratio (**d**) in roots (MELr and INSr) and leaves (MELl and INSl) in *Solanum melongena* (blue) and *S. insanum* (red), after 25 days of treatments with the indicated NaCl concentrations. Mean ± SE values are shown (*n* = 5). Same letters (lowercase for roots, or uppercase for leaves) indicate homogeneous groups between combinations of treatments, according to the Tukey HSD test (*p* < 0.05).

### 3.4. Osmolytes, MDA, and Antioxidants

A two-way ANOVA was performed to analyse the effects of the variables 'treatment' and 'species', as well as their interaction, on different biochemical parameters related to the general responses of plants to salt stress (Table 4). This analysis revealed a strong effect of 'treatment', but also a significant effect of 'species' and their interaction for proline. In the case of TSS, however, only the 'species' factor and its interaction with 'treatment' were significant. MDA showed a significant variation according to the treatment and the species; for total phenolic compounds (TPCs), the two factors and their interaction were significant, although the strongest contribution to the sums of squares was that of 'species'. For total flavonoids (TFs), the only significant effect was owing to the treatment. Regarding the antioxidant enzymatic activities, the two factors, treatment and species, as well as their interaction, were significant for SOD, whereas only the species effect was significant for CA, and no significant factor was detected for GR. It is remarkable that, for all biochemical compounds analysed, except proline, and for the three enzymatic activities, the percentage of the sum of square of residuals was the most important contributor to the sums of squares, indicating a high influence of uncontrolled residual variation (Table 4).

**Table 4.** Two-way analysis of variance (ANOVA) of treatment, species, and their interactions for the parameters considered. Numbers represent percentages of sum of squares (SS) at the 5% confidence level. Abbreviations: proline (Pro), total soluble sugars (TSSs), malondialdehyde (MDA), total phenolic compounds (TPCs), total flavonoids (TFs), superoxide dismutase (SOD), catalase (CAT), and glutathione reductase (GR).

| Trait | Treatment [a] | Species [a] | Interaction [a] | Residual |
|-------|-----------|---------|-------------|----------|
| Pro | 63.60 *** | 18.20 *** | 14.06 *** | 4.29 |
| TSS | 5.31 | 22.87 *** | 18.76 * | 53.05 |
| MDA | 25.29 *** | 29.23 *** | 6.48 | 38.98 |
| TPC | 8.19 * | 28.90 *** | 15.32 * | 47.57 |
| TF | 27.01 ** | 1.83 | 4.30 | 66.85 |
| SOD | 13.76 * | 21.11 *** | 16.92 * | 48.07 |
| CAT | 2.82 | 13.21 * | 10.26 | 73.24 |
| GR | 16.50 | 1.59 | 11.30 | 70.60 |

[a] ***, **, and * indicate significant at *p* < 0.001, *p* < 0.01, and *p* < 0.05, respectively.

Leaf proline (Pro) levels increased significantly in the two species in response to the salt stress treatments. In *S. melongena*, Pro contents were lower than in *S. insanum* at all tested salinities, reaching a peak in the presence of 200 mM NaCl, and decreasing at 300 mM NaCl. In *S. insanum*, Pro increased gradually in parallel to the external NaCl concentration, reaching levels about 10-fold higher than in the control at 300 mM NaCl (Figure 4a). Contrary to Pro, total soluble sugars (TSSs) in leaves showed a slight increase in salt-treated *S. melongena* plants, but the difference with the control was significant only in the presence of 200 mM NaCl. Average TSS contents were substantially higher in *S. insanum* than in *S. melongena* plants, in the control and at low salinity, to decrease at higher NaCl concentrations; however, the differences with the non-stressed controls were non-significant (Figure 4b).

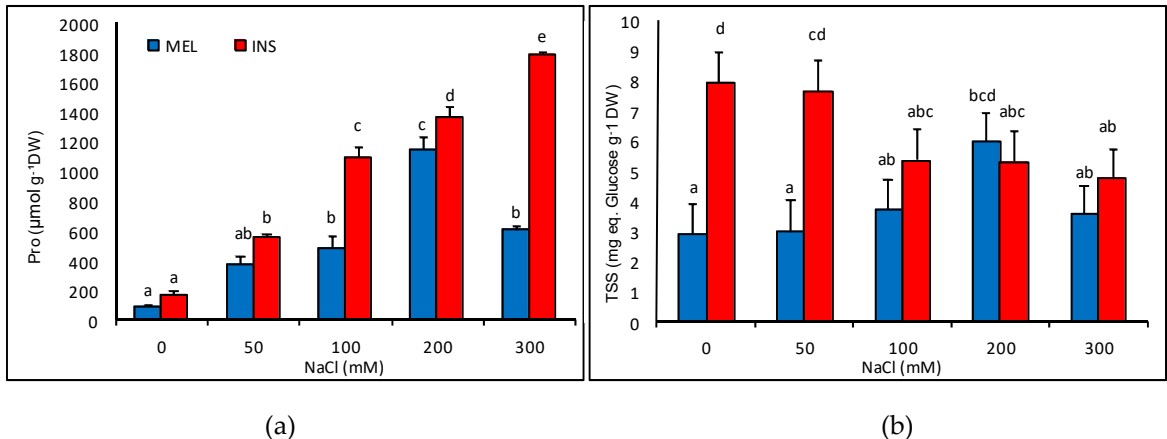

(a)　　　　　　　　　　　　　　　　　　　　　　　　　　(b)

**Figure 4.** Proline (Pro) (**a**) and total soluble sugars (TSSs) (**b**) contents in *Solanum melongena* (blue) and *S. insanum* (red) after 25 days of treatments with the indicated NaCl concentrations. Mean ± SE values are shown (*n* = 5). Same letters (lowercase for *S. melongena* and capital for *S. insanum*) indicate homogeneous groups between combinations of treatments, according to the Tukey HSD test (*p* < 0.05).

Malondialdehyde (MDA) is regarded as a reliable marker of oxidative stress, as it is a product of peroxidation of unsaturated fatty acids, indicating damage to cell membranes by 'reactive oxygen species' (ROS) in plants and animals [30]. However, its levels did not increase in salt-treated plants as compared with the controls, neither in *S. melongena* nor in *S. insanum*; on the contrary, leaf MDA contents slightly decreased in response to increasing salinity in plants of the two species (Table 5). A similar decreasing trend was observed for the mean values of the analysed antioxidant compounds, TPC and TF, although the differences with the non-stressed controls were not statistically significant in *S. melongena* (Table 5). Moreover, no significant salt-induced differences in specific activity could be detected in the assays of the antioxidant enzymes, SOD, CAT, and GR. When comparing the two species, higher MDA, TPC, and TF contents and higher specific enzyme activities were generally observed in *S. insanum*, at each external salt concentration tested (Table 5).

**Table 5.** Malondialdehyde (MDA), total phenolic compounds (TPCs), total flavonoids (TFs), and activity of the antioxidant enzymes: superoxide dismutase (SOD), catalase (CAT), and glutathione reductase (GR) in *S. melongena* (MEL) and *S. insanum* (INS) after 25 days of treatment with the indicated NaCl concentrations.

| | | Treatment (mM NaCl) | | | | |
|---|---|---|---|---|---|---|
| Trait | Taxa | 0 | 50 | 100 | 200 | 300 |
| MDA | MEL | 145.7 ± 12.2 [b] | 134.4 ± 9.1 [ab] | 114.7 ± 4.1 [ab] | 107.3 ± 7.5 [a] | 107.9 ± 7.8 [a] |
| | INS | 207.1 ±2 4.6 [B] | 143.9 ± 11.7 [A] | 145.3 ± 7.8 [A] | 145.2 ± 4.2 [A] | 162.1 ± 8.1 [AB] |
| TPC | MEL | 12.3 ± 0.7 [a] | 11.1 ± 2.1 [a] | 6.3 ± 0.6 [a] | 5.7 ± 0.4 [a] | 7.8 ± 1.3 [a] |
| | INS | 15.5 ± 0.8 [B] | 10.7 ± 1.8 [AB] | 7.3 ± 0.5 [A] | 6.3 ± 0.4 [A] | 10.6 ± 0.4 [AB] |

**Table 5.** *Cont.*

| Trait | Taxa | Treatment (mM NaCl) | | | | |
|---|---|---|---|---|---|---|
| | | 0 | 50 | 100 | 200 | 300 |
| TF | MEL | 9.5 ± 2.3 [a] | 7.8 ± 3.4 [a] | 9.1 ± 2.9 [a] | 5.0 ± 0.7 [a] | 6.0 ± 0.6 [a] |
| | INS | 15.5 ± 1.8 [B] | 10.5 ± 1.9 [AB] | 6.5 ± 0.9 [A] | 5.5 ± 0.6 [A] | 8.7 ± 0.7 [A] |
| SOD | MEL | 377.5 ± 46.9 [a] | 272.4 ± 24.2 [a] | 180.5 ± 30.8 [a] | 315.0 ± 20.3 [a] | 412.8 ± 75.0 [a] |
| | INS | 1181.0 ± 263.0 [A] | 464.9 ± 134.0 [A] | 915.0 ± 221.0 [A] | 586.3 ± 163.0 [A] | 315.6 ± 71.0 [A] |
| CAT | MEL | 280.2 ± 92.8 [a] | 453.4 ± 84.5 [a] | 304.3 ± 69.7 [a] | 514.1 ± 96.2 [a] | 413.2 ± 131.0 [a] |
| | INS | 1135.0 ± 416.0 [A] | 523.9 ± 80.4 [A] | 7214.0 ± 140.0 [A] | 560.8 ± 85.4 [A] | 692.1 ± 147 [A] |
| GR | MEL | 2419.0 ± 454.0 [a] | 1937.0 ± 384.0 [a] | 1468.0 ± 433.0 [a] | 1426.0 ± 268.0 [a] | 1484.0 ± 263.0 [a] |
| | INS | 2881 ± 684 [A] | 1523 ± 197 [A] | 3571 ± 764 [A] | 1390 ± 144 [A] | 1114 ± 77 [A] |

Units: MDA (nmol g$^{-1}$ DW), TPC (mg eq. GA g$^{-1}$ DW), TF (mg eq. C g$^{-1}$ DW), and enzymatic activity (U g$^{-1}$ protein). Mean ± SE values are shown (*n* = 5). Same letters within each row (lowercase for *S. melongena* and capital letters for *S. insanum*) indicate homogeneous groups between treatments for each species according to the Tukey HSD test ($p < 0.05$).

### 3.5. Principal Component Analysis

A principal component analysis (PCA) was performed, including all analysed traits in all individuals (Figure 5). Eight components with an Eigenvalue greater than one were identified, which overall explained 82.6% of the total variability; the first and second principal components accounted for 33.0% and 15.4% of the total variation, respectively. The first principal component displays positive correlations with growth parameters of stem (SE, ST, and SFW) and leaves (LA, LFW, and Lno), as well as with total fresh weight (TFW); carotenoids (Caro); photosynthetic parameters ($A_N$, Ci, E, gs); K in leaves (Kl); the ratio K/Na in roots (K/Nar) and leaves (K/Nal); as well as MDA, TP, and TF contents. On the other hand, this first PC is negatively correlated with the levels of $Na^+$ and $Cl^-$ in roots and leaves (Nar, Nal, Clr, Cll), and with Pro and root water content (RWC). The second component displays strong positive correlations with Pro, some photosynthesis parameters ($A_N$, E, gs), TSS, and CAT and SOD activities, whereas it is negatively correlated with root water content (RWC), leaf traits (LA, LFW, Lno), chlorophylls a and b (Chl a and Chl b), and $K^+$ contents in roots (Kr) and leaves (Kl) (Figure 5a).

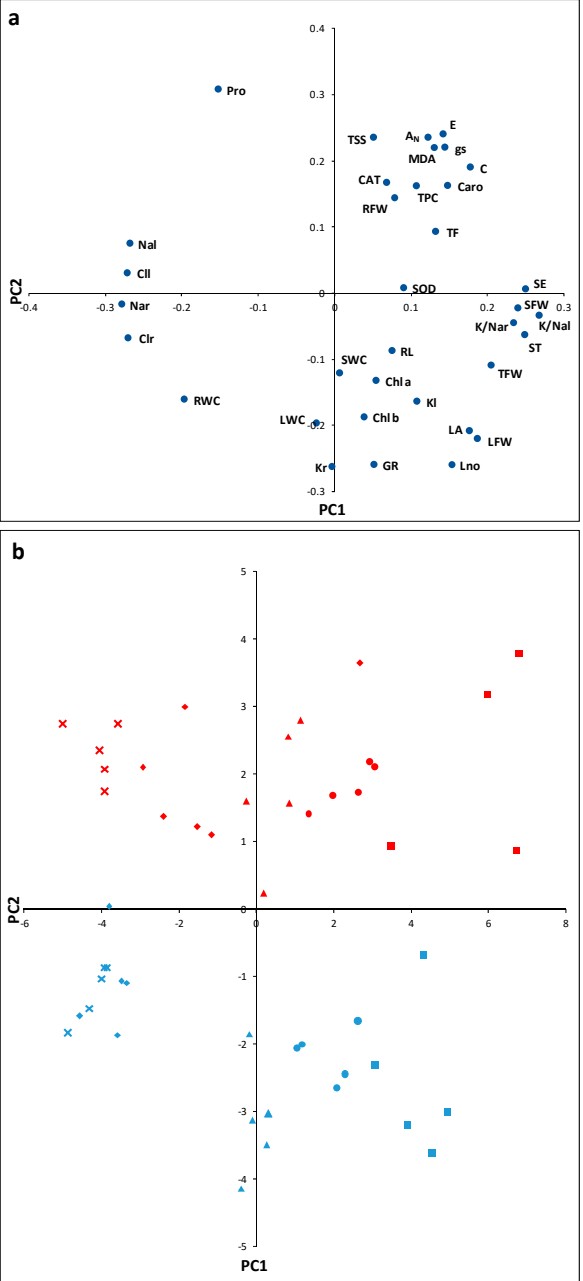

**Figure 5.** Loading plot (**a**) and scatterplot (**b**) of the principal component analysis (PCA) including all the analysed traits in *Solanum melongena* and *S. incanum* plants subjected for 25 days to salt treatments. The first (PC1; X-axis) and second (PC2; Y-axis) principal components accounted for 33.0% and 15.4% of the total variation, respectively. Abbreviations in the loading plot (**a**) are as follows: root length (RL), root fresh weight (RFW), root water content (RWC), stem elongation (SE), stem thickening (ST), stem fresh weight (SFW), stem water content (SWC), leaf number increment (Lno), maximal leaf area (LA), leaf fresh weight (LFW), leaf water content (LWC), total fresh weight (TFW), chlorophyll a (Chla), chlorophyll b (Chlb), carotenoids (Caro), photosynthetic rate ($A_N$), internal concentration of $CO_2$ (Ci), transpiration (E), stomatal conductance (gs), sodium in roots (Nar), sodium in leaves (Nal), potassium in roots (Kr), potassium in leaves (Kl), chloride in roots (Clr), chloride in leaves (Cll), ratio potassium/sodium in roots (K/Nar), ratio potassium/sodium in leaves (K/Nal), proline (Pro), total soluble sugars (TSS), malondialdehyde (MDA), total phenolic compounds (TPC), total flavonoids (TF), superoxide dismutase (SOD), catalase (CAT), and glutathione reductase (GR). Plants of *S. melongena* and of *S. insanum* are represented in blue and red, respectively, in the scatter plot (**b**). Salt treatments are represented by different symbols: 0 (■), 50 (●), 100 (▲), 200 (◆), and 300 (✕) mM NaCl.

The 50 individuals analysed were dispersed onto the two axis of the PCA scatterplot (Figure 5b), indicating a clear separation of the applied treatments along the first principal component (X-axis), and of the two species along the second principal component (Y-axis). Plants subjected to the different salt treatments are distributed along the X-axis, from higher positive values (non-stressed controls), to higher negative values (300 mM NaCl), with almost no overlapping of the different treatments, except for the 200 and 300 mM NaCl in *S. melongena*. Samples from moderate salinity treatments (50–100 mM NaCl for *S. melongena* and 100–200 mM NaCl for *S. insanum*) are located in the scatterplot in intermediate positions, closer to '0' in the X-axis. This pattern of distribution validates the homogeneity of responses within each treatment in the two species. Regarding the second principal component, except for one sample per species, *S. insanum* samples are located in the positive part of the Y-axis, whereas *S. melongena* samples have negative values for this component.

## 4. Discussion and Conclusions

Eggplant is a glycophyte and, as such, responds to increased salinity by a reduction in growth parameters and yield, being generally considered as moderately sensitive (or moderately resistant) to salt stress [7,17,31,32], as other cultivated species of the same genus [33]. However, this crop is characterised by a large variation of phenotypical, biochemical, and physiological traits, which is related to differences between cultivars in their responses to biotic [34] or abiotic stresses, including drought and salinity [35–37]. Therefore, the use of more stress-tolerant cultivars of eggplants on marginal lands or on salinised soils is a realistic challenge for the future, considering that global warming is generating an increased rate of secondary salinisation [38]. Soils are considered as saline when their EC (in a soil saturated paste) is above 4 dS m$^{-1}$; this electric conductivity corresponds to approximately 40 mM NaCl, generating an osmotic pressure of 0.2 MPa, which significantly reduces the yield of most crops [39]. These values cannot be directly compared with our results as we measured the substrate EC in soil/water (1:5) suspensions (EC$_{1:5}$), not in saturated soil pastes. Nevertheless, in our experiments, all concentrations of NaCl applied were higher than 40 mM, ranging from 50 to 300 mM NaCl. After 25 days of treatments, the salinity of the substrate in pots exposed to the higher concentrations of salt was clearly beyond that normally occurring on salinised soils. All plants survived the salt treatments, but, as expected, growth of stressed plants was reduced in comparison with those from the control treatments in the two investigated species, *S. melongena*, the cultivated eggplant, and its wild relative *S. insanum*.

The analysis of several growth parameters indicated that, in general, the degree of salt-induced growth inhibition was relatively lower in *S. insanum* than in *S. melongena*. One of the most reliable growth variables, when ranking stress tolerance in different cultivars or related species, is the variation of fresh weight (FW) of the plants [36,40,41]. The analysis of this parameter clearly indicated a better tolerance to high salinity in *S. insanum* as the FW of all vegetative organs (roots, stems, and leaves) showed a lesser reduction than in *S. melongena* in the presence of 200 mM and, especially, 300 mM NaCl. Under the 50 mM and 100 mM NaCl treatments, RFW and LFW even slightly increased in the wild species, indicating that these low concentrations have an inhibitory effect only on stem growth. A smaller increase, also non-significant, was registered under 50 mM NaCl for the leaf area (LA) and total fresh weight (TFW) in this species. The highest concentration of 300 mM was not lethal, as all individuals survived until the end of the experiment, but its effect was considerably stronger on *S. melongena*, as shown by a 60% reduction of the total fresh weight (TFW) as compared with only a 30% reduction in *S. insanum*. Special attention is required for the analysis of the root growth parameters because, apparently, all salt treatments stimulated root growth in *S. insanum*. On the contrary, although lower salt concentrations had a positive effect of root growth in *S. melongen*a, under the 300 mM NaCl treatment, root length (RL) and root fresh weight (RFW) were significantly reduced. Therefore, the development of more vigorous roots under salt stress represents an important adaptative trait in *S. insanum*. The water content (WC) of vegetative organs, particularly leaves, is another useful indicator of the relative salt tolerance of related taxa. The more tolerant species or cultivars are usually

resistant to salt-induced leaf dehydration, or at least the degree of water loss is lower than in the more sensitive ones [42,43]. Indeed, this has also been observed comparing different eggplant cultivars, with those more stress-tolerant showing higher leaf water contents under salt stress conditions [37]. It is worth mentioning that the specific eggplant cultivar used in the present work, MEL1, although more sensitive to salt stress than *S. insanum* INS2, is nevertheless quite tolerant to salinity, at least much more than other common crops such as *Phaseolus* cultivars [42]; all plants survived the salt treatments, even at 300 mM NaCl, and a significant growth inhibition was only observed at the highest salinities tested.

Salt stress reduces photosynthesis, which is one of the major reasons for growth inhibition [44,45]. One of the first effects of abiotic stress is the closure of stomata, which helps in reducing the water loss, but also limits the intake of $CO_2$. Therefore, in C3 plants (like the two species studied here), C assimilation decreases in such conditions [46]. The photosynthetic rate may also decrease owing to the degradation of chlorophylls or the inhibition of photosynthetic enzymes caused by toxic ions. The photosynthesis rate ($A_N$) decreased in the two species, but only in plants treated with the highest NaCl concentrations, not at lower salinities, as has been reported in different eggplant cultivars [47]. The internal concentration of $CO_2$ (Ci) and the transpiration (E) were reduced in *S. melongena* plants in response to the salt treatments, which is associated with a decrease in stomatal conductance (gs); this has also been observed in other cultivars of eggplant [48,49]. In *S. insanum*, however, salt stress did not induce any significant change in the above-mentioned photosynthetic parameters. On the other hand, in both species, chlorophylls a and b levels remained constant, for the control and all salt treatments, contrary to previous reports in eggplant [48,50]. The maintenance of a high assimilation rate in *S. insanum* may rely on its better developed root system, which allowed a higher water uptake under stressful conditions and a lower need for a restriction in transpiration (E), reflected in a higher stomatal conductance (gs) and internal concentration of $CO_2$ (C). Taken together, these results point to a slightly higher salt tolerance of *S. insanum* INS2, as compared with *S. melongena* MEL1.

Regarding ion accumulation, a significant increase in $Na^+$ and $Cl^-$ contents was registered in parallel to increasing external salinity, at 100 mM and higher NaCl concentrations, both in roots and leaves and in plants of the two species; similar results have been previously reported in different eggplant cultivars [7,48,49]. Glycophytes typically respond to salt stress trying to limit the accumulation of toxic ions in the leaves, either reducing their absorption by the roots or blocking their transport to the aerial parts of the plant [50]; these mechanisms are effective only at low or moderate salinities, and once a certain threshold—dependent on the tolerance of each specific genotype—is exceeded, $Na^+$ and $Cl^-$ concentrations increase in the leaves. In our experiments, no inhibition of $Na^+$ or $Cl^-$ transport from roots to leaves was observed because, generally, their concentration in roots was not higher than in leaves. In *S. melongena*, the concentration of the two ions was practically identical in roots and leaves, at each salinity level (except for $Na^+$ at 100 mM NaCl). Interestingly, in *S. insanum* plants treated with 200 or 300 mM NaCl, $Na^+$ concentrations in leaves were substantially higher than in roots, and the same pattern was observed for $Cl^-$ at 100 mM and higher NaCl concentrations. This suggests that, in this species, high salinity activates the transport of these ions from roots to leaves, where they could contribute to cellular osmotic balance as inorganic osmolytes. This is not a common behaviour of glycophytes like eggplant, but represents one of the most relevant mechanisms of salt tolerance in dicotyledonous halophytes [51,52], which could also be operative in *S. insanum*, contributing to its relative higher tolerance, enhanced also by a more developed root system that allows a higher ion uptake.

Potassium homeostasis is also critical for salt tolerance, which includes as a key mechanism the intracellular retention of $K^+$ in the presence of high external salinities [53,54], as this cation is essential in plant metabolism. An increase in $Na^+$ concentration is generally accompanied by a decrease of $K^+$, as both cations compete for the same membrane transport proteins [55]. Furthermore, high $Na^+$ levels produce a depolarisation of the plasma membrane, which induces $K^+$-efflux from cells by activating voltage-dependent outward rectifying channels [56,57]. Many reports indicated a reduction of $K^+$ in conditions of salt stress in eggplant, as expected [32,48,49]. In our experiments, however, no significant

changes in root or leaf $K^+$ concentrations were observed in response to the salt treatments. Maintenance of constant $K^+$ levels, despite the increase in $Na^+$ concentrations, probably also contributes to salt tolerance, in this case, in both tested genotypes, *S. melongena* MEL1 and *S. insanum* INS2. Further studies will be required to elucidate the specific ion transporters involved in these regulatory mechanisms.

Another general response to salt stress is the synthesis of Pro, one the commonest osmolytes in plants, which, besides osmotic adjustment, plays an important role in ROS detoxification and maintenance of membrane integrity under stress [58,59]. Pro accumulation may be simply a biomarker of the level of stress affecting a plant, reaching higher concentrations in the more stressed individuals, as has been shown in some comparative studies on related genotypes [42]. On the contrary, Pro can be directly involved in the mechanisms of tolerance to stress, so that higher contents correlate with higher tolerance [40]. Comparative analyses of different eggplant cultivars have provided mixed results; in some cases, the more stress-tolerant genotypes accumulated higher Pro concentrations [35,36,60], but in other studies, higher levels were found in the more sensitive ones [32]. Our results clearly showed higher Pro levels in *S. insanum* than in *S. melongena* in all experimental conditions, but especially in the presence of the highest salinity tested, 300 mM NaCl, thus correlating with the relative salt tolerance of the two investigated species.

Although soluble sugars play a role in osmoregulation under stress conditions in many plant species [61], their levels did not vary significantly in response to the salt treatments in *S. insanum*, and were similar in the two species at high salinities. Therefore, TSS contents do not correlate with the degree of salt tolerance, and probably do not play any relevant role in the responses to salt stress of the two species studied here.

Mechanisms of salt tolerance based mostly on the accumulation of Pro, for osmotic adjustment and as 'osmoprotector'—with the possible contribution of $Na^+$ and $Cl^-$ as inorganic osmolytes in the case of *S. insanum*—appear to be efficient enough to avoid the generation of oxidative stress under the specific conditions used in our experiments. A common effect of high salinity, as well as other abiotic stresses, is the increase in the concentration of ROS, leading to secondary oxidative stress [62]. That did not occur in the present work, as shown by the determination of MDA contents, which did not increase in response to the salt treatments. Consequently, the activation of antioxidant systems, enzymatic and non-enzymatic, was also not detected, as the plants did not need to counteract any salt-induced oxidative stress. Generally, this behaviour is not observed in glycophytes, but has been reported for many halophytes [26,63].

In conclusion, our results from plant growth, photosynthetic parameters, and biochemical stress markers measurements indicate that *S. insanum* displays greater tolerance to moderate salt stress than *S. melongena*, mostly because of its ability to accumulate higher concentrations of Pro and, to a lesser extent, $Na^+$ and $Cl^-$ in the leaves, especially at high external salinities. Given that *S. insanum* and *S. melongena* are fully cross-compatible [10,16], and introgression breeding from *S. insanum* into *S. melongena* is relatively easy [11], we suggest that *S. insanum* can contribute to the development of *S. melongena* cultivars with increased salt tolerance. It remains to be evaluated if *S. insanum* could also be useful as a rootstock for eggplant under conditions of salinity. Therefore, the use of *S. insanum* in eggplant breeding and rootstock development may make an effective contribution to extending cultivation of eggplant in cultivated lands that are affected, or will be in the future, by soil salinity.

**Author Contributions:** Conceptualization, O.V. and J.P.; Data curation, M.B. (Marco Brenes) and M.B. (Monica Boscaiu); Formal analysis, J.P.; Funding acquisition, O.V. and J.P.; Investigation, M.P.; Methodology, M.B. (Marco Brenes) and A.S.; Project administration, O.V.; Resources, O.V. and J.P.; Software, M.B. (Monica Boscaiu); Supervision, M.P.; Validation, A.F. and A.C.; Visualization, M.B. (Monica Boscaiu) and M.P.; Writing—original draft, M.B. (Monica Boscaiu) and J.P.; Writing—review & editing, A.F., O.V., A.C., and M.P. All authors have read and agreed to the published version of the manuscript

**Funding:** This work was undertaken as part of the initiative "Adapting Agriculture to Climate Change: Collecting, Protecting and Preparing Crop Wild Relatives", which is supported by the Government of Norway and managed by the Global Crop Diversity Trust. For further information, see the project website: http://cwrdiversity.org/. Funding was also received from Ministerio de Ciencia, Innovación y Universidades, Agencia Estatal de Investigación and Fondo Europeo de Desarrollo Regional (grant RTI-2018-094592-B-100 from MCIU/AEI/FEDER, UE), European

Union's Horizon 2020 Research and Innovation Programme under grant agreement No. 677379 (Linking genetic resources, genomes, and phenotypes of Solanaceous crops; G2P-SOL) and Vicerrectorado de Investigación, Innovación y Transferencia de la Universitat Politècnica de València (Ayuda a Primeros Proyectos de Investigación; PAID-06-18). Mariola Plazas is grateful to Generalitat Valenciana and Fondo Social Europeo for a post-doctoral grant (APOSTD/2018/014). Marco Brenes is indebted to the Faculty of Biology of the Costa Rica Institute of Technology for partially supporting his stay in Valencia ("Fondo Solidario y Desarrollo Estudiantil").

**Conflicts of Interest:** The authors declare no conflict of interest. The funders had no role in the design of the study; in the collection, analyses, or interpretation of data; in the writing of the manuscript; or in the decision to publish the results.

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
