# Peer review of "Physiological and Biochemical Responses to Salt Stress in Cultivated Eggplant (Solanum melongena L.) and in S. insanum L., a Close Wild Relative"

_agronomy, doi:10.3390/agronomy10050651_

Round 1

Reviewer 1 Report

The research aimed in answering the question whether Solanum insanum, a wild egg plant relative, can be a suitable donor of salt tolerance that could be transferred then by hybridization with cultivated forms of this vegetable. The manuscript presents results of physiological and biochemical response of two egg plant accessions to salt stress, a cultivated MEL1 accession and S. insanum (INS). Several parameters related to biomass growth, water content, photosynthesis and activity of antioxidant system were measured in plants of both accessions exposed to a range of NaCl concentrations applied by plant watering. In general, the experimental design is correct although five plants used per treatment per accessions rise my concern as in this layout one plant represents one biological replication. Moreover there is no information how, and how many technical replications were prepared for physiological and biochemical measurements and whether they were used to calculate mean values used then for analysis of variance or they all were included in those analyses.

Despite this methodological concern, I would suggest to provide information why MEL1 accessions was chosen. Is it a good representative of cultivated egg plant , is it commonly grown ?

In general, the manuscripts shows only moderate differences between both accessions in their reaction to increasing level of salt stress. Thus the Authors indicate that they differ and less adverse effects are seen in S. insanum plants, but actually they cannot explain the mechanisms that are responsible for that as the changes in physiological parameters are small.  Despite that the knowledge that  S. insanum may have higher tolerance to salinity is worth publication. I would suggest however the Authors focus more on how both accessions react to increasing levels of applied NaCl in particular whether they see stimulating effect of low NaCl concentrations and whether final decrease of some parameters at the highest dose may be an effect of cell/tissue death.

I would also suggest to extend the discussion on photosynthesis (page 14, lines 445-459) and in particular its conclusion to be more related to the preceding statements. In my oppinion, the discussion should start from the observation that INS has a better developer root system than MEL, which may explain a lot, higher water uptake and in consequence higher uptake and accumulation of Na and Cl, lower need for restriction in transpiration  etc.. leading in consequence to higher assimilation rate than in case of MEL.  Thus my question is, whether the observed higher tolerance on INS to salinity can be due to different plant architecture (larger root system) rather to physiological response to stress.

For clarity and better understanding, the list of traits assessed and analysed by ANOVA should contain also Total Fresh Weight (Total biomass) of plants.

Figure 2 is misleading. From Table 2 it is clear that leaf fresh weight of the control MEL plants is almost doubled in comparison to control INS plants, but looking at Fig. 2 the impression is opposite.

Table 1 – insert a new column at left side with full names of the traits. Units given in the Table header are not applicable here as the numbers in the table show percentage values of the sum of squares and not mean values.

Author Response

The response to the editor and referees’ comments and the changes made in the manuscript are indicated below. Reviewers’ comments are listed and our responses are preceded by an asterisk (*).

The research aimed in answering the question whether Solanum insanum, a wild egg plant relative, can be a suitable donor of salt tolerance that could be transferred then by hybridization with cultivated forms of this vegetable. The manuscript presents results of physiological and biochemical response of two egg plant accessions to salt stress, a cultivated MEL1 accession and S. insanum (INS). Several parameters related to biomass growth, water content, photosynthesis and activity of antioxidant system were measured in plants of both accessions exposed to a range of NaCl concentrations applied by plant watering. In general, the experimental design is correct although five plants used per treatment per accessions rise my concern as in this layout one plant represents one biological replication. Moreover there is no information how, and how many technical replications were prepared for physiological and biochemical measurements and whether they were used to calculate mean values used then for analysis of variance or they all were included in those analyses.

(*) Our response: In subsection 2.1. Plant material we have indicated that each plant constituted a biological replica: “Five plants of each species, each one corresponding to a biological replica,…”. This experimental design has been validated previously for tolerance to drought in eggplant (Plazas et al., 2019) and allows sufficient statistical power to detect relevant differences among treatments. We have also indicated that “Measurements for physiological, biochemical and ion content parameters were based in one technical replicate”. As we included detailed information on the experimental layout in this subsection, we have renamed it as “2.1. Plant Material and Experimental Layout”.

Despite this methodological concern, I would suggest to provide information why MEL1 accessions was chosen. Is it a good representative of cultivated egg plant , is it commonly grown ?

(*) Our response: We have included the reason for selecting accession MEL1 in Material and Methods subsection 2.1. Plant Material and Experimental Layout: “Solanum melongena MEL1 was chosen since this accession is of particular interest for breeding as it has an excellent fruit set and shows a high degree of success in interspecific hybridisation [10,11].”

In general, the manuscripts shows only moderate differences between both accessions in their reaction to increasing level of salt stress. Thus the Authors indicate that they differ and less adverse effects are seen in S. insanum plants, but actually they cannot explain the mechanisms that are responsible for that as the changes in physiological parameters are small. Despite that the knowledge that S. insanum may have higher tolerance to salinity is worth publication. I would suggest however the Authors focus more on how both accessions react to increasing levels of applied NaCl in particular whether they see stimulating effect of low NaCl concentrations and whether final decrease of some parameters at the highest dose may be an effect of cell/tissue death.

(*) Our response: Thank you very much for this suggestion. At low NaCl concentrations growth of S. insanum is enhanced, although the differences are not statistically significant. However, it is clear that under low and intermediate salt concentrations fresh weight is increasing in the wild species, excepting that of the stem. Tissue death was not registered, and all plants of the two species survived the treatments, but percentages of reduction under the strongest salt concentrations were considerably higher in S. melongena than in S. insanum. An explanatory paragraph has been introduced in the Discussion: “Under the 50 mM and 100 mM NaCl treatments, RFW and LFW even slightly increased in the wild species, indicating that these low concentrations have an inhibitory effect only on stem growth. A smaller increase, also non-significant, was registered under 50 mM NaCl for the leaf area (LA) and total fresh weight (TFW) in this species. The highest concentration of 300 mM was not lethal, as all individuals survived until the end of the experiment, but its effect were considerably stronger on S. melongena, as shown by a 60% reduction of the total fresh weight (TFW) as compared to only 30% reduction in S. insanum. A special attention needs the analysis of the root growth parameters, as apparently all salt treatments stimulated root growth in S. insanum. On the contrary, although lower salt concentrations had a positive effect of root growth in S. melongena, under the 300 mM NaCl treatment root length (RL) and root fresh weight (RFW) were significantly reduced. Therefore, the development of more vigorous roots under salt stress represents an important adaptative trait in S. insanum.

I would also suggest to extend the discussion on photosynthesis (page 14, lines 445-459) and in particular its conclusion to be more related to the preceding statements. In my oppinion, the discussion should start from the observation that INS has a better developer root system than MEL, which may explain a lot, higher water uptake and in consequence higher uptake and accumulation of Na and Cl, lower need for restriction in transpiration  etc.. leading in consequence to higher assimilation rate than in case of MEL.  Thus my question is, whether the observed higher tolerance on INS to salinity can be due to different plant architecture (larger root system) rather to physiological response to stress.

(*) Our response: Thank you very much for this remark. We followed your suggestion and emphasised the connection between the differences in the development of root systems under stress and that registered in the photosynthetic parameters and ion absorption by adding the following text in the Discussion: “The maintenance of a high assimilation rate in S. insanum may rely on its better developed root system, which allowed a higher water uptake under stressful conditions and a lower need for a restriction in transpiration (E), reflected in a higher stomatal conductance (gs) and internal concentration of CO2 (C).”

For clarity and better understanding, the list of traits assessed and analysed by ANOVA should contain also Total Fresh Weight (Total biomass) of plants.

(*) Our response: The total fresh weight (TFW) has been included as a new parameter in the two corresponding ANOVA tables and also in the PCA. Including this trait reveals better the differences between the two species than in the previous version when fresh weight was considered only separately for roots, stems and leaves. A new paragraph on TFW was included in the Results section, and the parameter is also mentioned in the Discussion “When considering the total fresh weight (TFW), the reduction was significant only in the cultivated eggplant, but not in the wild species, in which at the lowest concentration applied TFW even increased, but the variation was not statistically significant in relation to the control.”

Figure 2 is misleading. From Table 2 it is clear that leaf fresh weight of the control MEL plants is almost doubled in comparison to control INS plants, but looking at Fig. 2 the impression is opposite.

            (*) Our response: The reviewer is that right in pointing the fact that the figure 2 may be misleading. Therefore, we have removed it from the manuscript.

Table 1 – insert a new column at left side with full names of the traits. Units given in the Table header are not applicable here as the numbers in the table show percentage values of the sum of squares and not mean values.

 (*) Our response: We completely agree. Units were deleted and a new column with full names of traits has been included.

Reviewer 2 Report

Manuscript ID: agronomy-775367
Type of manuscript: Article
Title: Physiological and biochemical responses to salt stress in cultivated
eggplant (Solanum melongena L.) and in S. insanum L., a close wild relative

The objective of this study was determining the level of  tolerance to salinity of a wild eggplant genotype of S. insanum, wtih S. melongena by analysing the variation of growth  traits, photosynthesis and biochemical responses associated with tolerance to salinity such as: levels  of ions accumulated in different tissues, osmolytes, and antioxidants.

The data have some merit for possible publication in Agronomy but the manuscript needs revision.

  1. Abstract is quite large, make it shorter keeping the key results. Present in correct grammar for the international readers.
  2. Label every pot in figure 2a and 2b.
  3. In figure3, why fresh weight of roots of MEL is greatly reduced but that of leaves and stems were not under 300 mM NaCl?
  4. Line 85: what were the greenhouse conditions?
  5. Line 87-90: Any irrigation given at all to 0 mM NaCl treatment?
  6. Line 137: How many biological replica did you have for each variable?
  7. Line 193-94: Conduct a two-way ANOVA to investigate genotype, treatment and genotype x treatment interaction. Posthoc analysis should represent the interaction effect in the figures. Why did you put letters of individual genotype only?
  8. Line 205: S. melongena italic
  9. Line 265-67: Statement is not consistent with figure 3.
  10. Line 370-380: PCA cannot be commented because Figure 6 and 7 are not labelled with PC1 and PC2. Apparently looks like some statements are not valid.
  11. Combine Figure 6 and Figure 7 (PCA). Indicate PC1 and PC2 in figure.
  12. Line 427-28: Have you quantified the growth of the plants after treatment?
  13. Line 473: activates italic?
  14. Lines 550-59: where are the appendices and supplementary data?

Author Response

The response to the editor and referees’ comments and the changes made in the manuscript are indicated below. Reviewers’ comments are listed and our responses are preceded by an asterisk (*).

The objective of this study was determining the level of  tolerance to salinity of a wild eggplant genotype of S. insanum, wtih S. melongena by analysing the variation of growth  traits, photosynthesis and biochemical responses associated with tolerance to salinity such as: levels  of ions accumulated in different tissues, osmolytes, and antioxidants.

The data have some merit for possible publication in Agronomy but the manuscript needs revision.

  1. Abstract is quite large, make it shorter keeping the key results. Present in correct grammar for the international readers.

(*) Our response: The Abstract has been reduced from 302 to 230 words, keeping the relevant information. The grammar of the Abstract has also been reviewed.

  1. Label every pot in figure 2a and 2b.

(*) Our response: Figure 2 has been removed, according to the suggestion of reviewer #1, it could be misleading.

  1. In figure3, why fresh weight of roots of MEL is greatly reduced but that of leaves and stems were not under 300 mM NaCl?

(*) Our response: Figure 3 included an error. We apologise for this mistake. A corrected new figure was included (Figure 2 in the revised version).

  1. Line 85: what were the greenhouse conditions?

(*) Our response: We have indicated in subsection 2.1. Plant Material and Experimental Layout that the “The plants were transferred to a greenhouse with benches and controlled temperature (maximum of 30 ºC and minimum of 15 ºC)”

  1. Line 87-90: Any irrigation given at all to 0 mM NaCl treatment?

(*) Our response: Control plants received irrigation with deionised water with the same frequency and the same volumes as plants from the stress treatments. The sentence has been reformulated and now the irrigation procedure is more clearly described.

Line 137: How many biological replica did you have for each variable?

(*) Our response: For all parameters the number of biological replicas (plants) was 5, as it is indicated in the legends of the tables and figures, and also in subsection 2.1. Plant Material and Experimental Layout: “Five plants of each species, each one corresponding to a biological replica,…”

  1. Line 193-94: Conduct a two-way ANOVA to investigate genotype, treatment and genotype x treatment interaction. Posthoc analysis should represent the interaction effect in the figures. Why did you put letters of individual genotype only?

(*) Our response: Data of the two-way ANOVA are shown in tables 1, 3 and 4. Figures are based on mean values for each treatment and each species and letters indicate homogenous groups separately for each species.  In the present version we follow your suggestion and now pairwise comparison between any combination of species and salinity treatment can be made.

  1. Line 205: S. melongena italic

(*) Our response: Corrected.

  1. Line 265-67: Statement is not consistent with figure 3.

(*) Our response: This is true because Figure 3 (Figure 2 in the revised version) included an error, which we have corrected. Thank you for noticing this error.

  1. Line 370-380: PCA cannot be commented because Figure 6 and 7 are not labelled with PC1 and PC2. Apparently looks like some statements are not valid.

(*) Our response: We have indicated in the legend of the figure that the x-axis corresponds to the first principal component while the y-axis to the second component: “The first (x-axis) and second (y-axis) principal components account, respectively, for 33.0% and 15.4% of the total variation”.

  1. Combine Figure 6 and Figure 7 (PCA). Indicate PC1 and PC2 in figure.

(*) Our response: The figures were combined in a single one (Figure 5 in the revised version). We have indicated in the figure legend that the x-axis corresponds to the first principal component while the y-axis to the second component.

  1. Line 427-28: Have you quantified the growth of the plants after treatment?

(*) Our response: Non-destructive growth parameters such as stem length, stem diameter and number of leaves were checked at the beginning of the treatments; so values reported represent their variation. For all other parameters final values, after the treatments, are given. To avoid confusion, we specify “growth of stressed plants was reduced in comparison to those from the control treatment.”

  1. Line 473: activates italic?

(*) Our response: Corrected to normal font.

  1. Lines 550-59: where are the appendices and supplementary data?

 (*) Our response: There are no appendices or supplementary data in the manuscript. This section belongs to the template facilitated by the editorial office. In consequence, it has been removed.

Round 2

Reviewer 1 Report

The Authors introduced the suggested changes that are satisfactory.  I have no more comments. 

Author Response

The Authors introduced the suggested changes that are satisfactory.  I have no more comments. 

(*) Our response: Thanks for your positive comments.

Reviewer 2 Report

  1. New Figure 5: Indicate which one is PC1 (in which axis) and which one is PC2 in the figure. It is possible to combine loading plot and score plot in one figure (not A and B). Some software can do. All variables in the loading plot are not defined. What are RL, K/Nal and K/Nar etc.?
  2. Separation of coefficients was not carefully done across PC1 and PC2. For example, what was the contribution of K/Nal in PC1? K/Nal has high (may be the highest) positive coefficient.
  3. New figure 3: Roots and leaves should be presented in distinct colour as data were analyzed (lettered) separately.

Author Response

  1. New Figure 5: Indicate which one is PC1 (in which axis) and which one is PC2 in the figure. It is possible to combine loading plot and score plot in one figure (not A and B). Some software can do. All variables in the loading plot are not defined. What are RL, K/Nal and K/Nar etc.?

(*) Our response: In the reviewed version it was indicated in the legend that the PC1 corresponds to the X-axis, while PC2 to the Y-axis. In any case, as suggested by the reviewer, we have included in the figure proper, which axis is PC1 and which one is PC2. Regarding the combination of the loading plot and score plot in a single biplot figure, due to the great number of points in each of the graphs the result is a messy figure in which it is difficult to ascertain either the traits or the data points. Therefore, we consider that it is much better for a clearer interpretation by readers of having a separation in the same figure of the loading plot and the score plot of the PCA. We are thankful to the reviewer for having spotted the traits [sodium in roots (Nar), sodium in leaves (Nal), potassium in roots (Kr), potassium in leaves (Kl), chloride in roots (Clr), chloride in leaves (Cll), ratio potassium-sodium in roots (K/Nar), ratio potassium-sodium in leaves (K/Nal)] that were not defined in the figure legend. Now, we have include these definitions.

2.  Separation of coefficients was not carefully done across PC1 and PC2. For example, what was the contribution of K/Nal in PC1? K/Nal has high (may be the highest) positive coefficient.

(*) Our response: We thanks the reviewer for having spotted this incomplete listing of traits contributing strongly to PC1 and PC2. We have corrected the corresponding paragraph, including the traits with most relevant contribution to the PC1 and PC2.

3. New figure 3: Roots and leaves should be presented in distinct colour as data were analyzed (lettered) separately.

(*) Our response: This observation makes a lot of sense. According to the suggestion of the reviewer we have indicated in blue the columns corresponding to roots (pale blue for S. melongena and dark blue for S. insanum) and leaves (pale red for S. melongena and dark blue for S. insanum).